# Research on Total Factor Productivity and Influential Factors of the Regional Water–Energy–Food Nexus: A Case Study on Inner Mongolia, China

**DOI:** 10.3390/ijerph16173051

**Published:** 2019-08-22

**Authors:** Junfei Chen, Tonghui Ding, Huimin Wang, Xiaoya Yu

**Affiliations:** 1State Key Laboratory of Hydrology-Water Resources and Hydraulic Engineering, Hohai University, Nanjing 210098, China; 2Business School, Hohai University, Nanjing 211100, China

**Keywords:** total factor productivity, influential factors, super-efficiency SBM model, Malmquist–Luenberger index, water–energy–food nexus, Inner Mongolia

## Abstract

With the supply of water, energy and food facing severe challenges, there has been an increased recognition of the importance of studying the regional water–energy–food nexus. In this paper, Inner Mongolia, including 12 cities in China, was selected as a research case. A super-efficiency slack based measure (SBM) model that considered the undesirable outputs was adopted to calculate the regional total factor productivity (TFP) and the Malmquist–Luenberger index was used to investigate the change trend of the TFP from 2007 to 2016 based on understanding the water–energy–food nexus. Finally, influential factors of the TFP were explored by Tobit regression. The results show that the 12 Inner Mongolia cities are divided into higher, moderate and lower efficiency zones. The higher efficiency zone includes Ordos, Hohhot, Xing’an, and Tongliao, and the lower efficiency zone includes Chifeng, Xilin Gol, Baynnur, Wuhai and Alxa. There is a serious difference in TFP between Inner Mongolia cities. During the study period, the TFP of the water–energy–food nexus in Inner Mongolia cities shows a rising trend, which is mainly driven by the growth of technical progress change. However, the average ML values of the lower and moderate efficiency zones were inferior to the higher efficiency zone in six of the ten years, so the difference between Inner Mongolia cities is growing. According to the Tobit regression, the mechanization level and degree of opening up have positive effects on the TFP, while enterprise scale and the output of the third industry have negative effects on the TFP. Government support does not have any significant impact on the TFP. Finally, suggestions were put forward to improve the TFP of the water–energy–food nexus in Inner Mongolia cities.

## 1. Introduction

Water, energy and food are important resources for economic and social development [1]. However, it is estimated that the global demand for energy and food will increase by 50% and water resources by 30% by 2030, while the supply of water, energy and food will face severe challenges due to the impact of ecological change. The imbalance between the supply and demand of the water–energy–food (WEF) nexus will become a serious problem. Therefore, a highly sensitive and fragile relationship has gradually formed between water, energy and food [2]. The study on the WEF nexus has gradually caught the attention of scholars and has become a hot topic in recent years [3].

In January 2011, the global risk report put forward the “water resources–energy–food risk group” as one of the three key risk groups, which emphasized that the policy of optimizing a single resource would cause unpredictable consequences. We should grasp and give attention to the production and consumption of the regional WEF nexus as a whole [4]. In November 2011, the Bonn conference firstly summarized the relationship between water, energy and food as a bond, and actively tried to find ways to balance the synergies between water, energy and food using a nexus method. The conference proposed improving the utilization efficiency of water, energy and food by reducing transactions and improving the governance of various sectors so as to achieve a green economy [5,6,7,8]. In 2013, the WEF nexus report in the Asia-pacific region pointed out that the WEF nexus was closely related in time and space. The report thought that the influential external factors of the WEF nexus included climate change, population growth, accelerating urbanization, etc., in the Asia-pacific region [9].

There is growing concern from numerous researchers regarding the WEF nexus. Academia especially focuses on the definition and challenges of the WEF nexus. Hoff [5] introduced the definition of the WEF nexus and thought that it could enhance the synergy of the three core resources in production and consumption and improve the utilization efficiency. Amorim [10] gave the definition and analyzed influential factors of the WEF nexus through literature and qualitative research. Fan et al. [11] used a multiple linear regression in a simultaneous equation model (SEM) to conduct the integrated evaluation of the WEF nexus and identified the key factors in the WEF nexus. Nhamo et al. [12] thought that the WEF nexus was a conceptual framework that presented opportunities for greater resource coordination, management, and policy convergence across sectors. Villamayor-Tomas et al. [13] examined the WEF nexus under an institutional development framework. Li et al. [2] used system dynamics to research the WEF nexus quantitatively. Bazilian et al. [14] stressed the importance of disposing the WEF nexus globally by building integrated models. Kurian [15] provided the framework of the WEF nexus in the study and concentrated on the importance of an interdisciplinary approach. Pahl-Wostl [16] defined WEF security from the perspective of the WEF nexus, and introduced an analytical framework based on a network and ecological service system. Foran [17] posited that production and consumption of WEF had an impact on economic, social and environmental subsystems in the region. Karnib [18] presented a generic scenario-based framework of using the Q-Nexus Model for informing about the nexus effects that needed to be reflected in WEF planning and policy-making settings. Zhang et al. [19] believed that the research on the WEF nexus is advanced, but there are still limitations. In the future, the WEF nexus faces four major challenges, namely the definition of the system boundary, the uncertainty associated with modeling, restrictions on the internal mechanism, and the evaluation of system performance. Xu et al. [20] studied the coupling and coordination degrees of the WEF nexus and found key factors that affected the development levels of WEF. In terms of the research method of the WEF nexus, most of the references mainly adopted qualitative methods, such as an entity model evaluation method. Some scholars have tried to simulate the WEF nexus quantitatively with the method of system dynamics and complex network analysis and have made progress. Halbe et al. [21] analyzed the core relationship of the WEF nexus by using a causality diagram, which provided a foundation for the construction of a subsequent system dynamics model. Oz et al. [22] simulated the water–energy–climate correlation using the system dynamics model in a study on the WEF nexus. Li et al. [2] used the system dynamics simulation technology to simulate the dynamics of the WEF nexus in Beijing and carried out the sensitivity analysis of the WEF nexus. Therefore, this paper studies the total factor productivity (TFP) of the regional WEF nexus based on an understanding of the WEF nexus.

At present, studies on the contribution of the consumption of WEF to the regional economy and environment are quite rare. Most scholars have considered the efficiency of a single resource or two resources rather than the overall efficiency of the WEF nexus. In terms of the efficiency of a single resource, Rad et al. [23] measured water efficiency and productivity with the aim of reducing water use in metropolises based on data envelopment analysis (DEA). Ding [24] adopted the super-efficiency slack based measure (SBM) model considering undesirable outputs to measure the water utilization efficiency of the Yangtze River. Zhang et al. [25] constructed the related random effect model to reduce the endogenous problem of traditional production function and analyzed the effect of nitrogen application on the food production efficiency at the household level. Ouyang et al. [26] adopted stochastic frontier analysis (SFA) to evaluate the industrial total factor energy efficiency of nine cities in Pearl River Delta (PRD), and then decomposed the growth of total factor energy efficiency into technical progress change and technical efficiency change. Borozan et al. [27] explored the total-factor energy efficiency and change trends in technical efficiency in Croatian counties based on data envelopment analysis and assessed the overall technical, pure technical and scale efficiency of energy. Duan [28] studied the efficiency of water for food production in Jilin province by using GIS spatial analysis and DEA and analyzed the main influential factors of local food production water footprint. Dong [29] used the improved SFA model to empirically decompose the production efficiency of six categories of food varieties in China from the perspective of energy and further analyzed the influential factors of food production efficiency. In terms of the comprehensive evaluation of water resources, energy and food, Ozturk [30] used a dynamic panel estimation and principal component analysis to study the sustainable development of the WEF nexus in BRICS (Brazil, the Russian Federation, India, China, and South Africa). Vito [31] proposed an integrated evaluation index system for the sustainability of irrigation based on the analysis of the bond between water, energy and food. It can be observed that there is lack of research on the overall efficiency of regional WEF.

The TFP comes from the development of productivity theory, which goes through single-factor productivity and the TFP. Single-factor productivity was produced along with classical economic theory, only studying the influence of a single input on output, such as capital productivity, labor productivity and ground productivity, etc, which is difficult to fully determine the comprehensive level of productivity. [32] However, due to the fact that capital, labor and other inputs are often used in the production process at the same time, there are usually substitutions among different production factors. The TFP, therefore, was built based on neoclassical economic theory and endogenous economic growth theory. Davis [33] firstly elaborated the connotation of the TFP in his book, which was understood as the overall efficiency of transforming all inputs into various outputs. Then Aigner et al. [34] divided the TFP into technical efficiency change and production possibility boundary change, which can be used to study the root of economic growth. There is some research about the applications of the TFP. Early research mostly applied this method to evaluate the TFP. For example, Zhong et al. [35] calculated Chinese agricultural TFP by using DEA and building a spatial panel data model embedded with climate change factors and then explored the possible impact of climate change on the regional agricultural TFP at a provincial level in China. Maryam et al. [36] investigated the principal determinants of TFP convergence by employing data of 91 developing countries over the period 1960–2015, with the USA being the frontier country. Danska-Borsiak [37] estimated the TFP for 35 NUTS 2 regions of the Visegrad Group countries and studied the parameters of the productivity function using a fixed effects model. In summary, considering all inputs and undesired outputs, the TFP is the comprehensive reflection of input–output efficiency, which is the embodiment of regional social and economic development. The growth of the TFP is usually attributed to the improvement in technology and management ability. Therefore, in this paper, we calculated the TFP of regional WEF to observe economic and environmental development. Data envelopment analysis (DEA) is generally used to evaluate multiple orientation and multiple decision-making priorities through the establishment of the efficiency index including input and output variables for each decision-making unit (DMU). DEA and the relevant models have experienced a long process of development. Firstly, Farrell [38] put forward the concepts of DEA. Later, Charnes [39] and Banker [40] built the Charnes, Cooper, and Rhodes (CCR) model and the Banker, Charnes, and Cooper (BCC) model respectively. Then Tone [41] put forward the SBM model to solve the problem that the DMU could not be estimated in the case of variable returns to scale. Andersen and Petersen [42] raised the super-efficiency model to resolve the problems when a super efficiency value ≥1 appeared. Song et al. [43] proposed a variant of the super-efficiency DEA model considering undesirable outputs and calculated the super-efficiency values to realize the complete ranking of all DMUs. Gokgoz et al. [44] calculated the energy efficiency of European countries and regions for the period of 2011–2015 using super efficiency and super SBM models. In this paper, when studying the TFP of the regional WEF nexus, not only does it produce desirable outputs such as Gross domestic product (GDP), but it also produces undesirable outputs such as waste gas. Therefore, we adopted the super-efficiency SBM model to calculate the TFP, so that we can rank all DMUs through calculation results. In addition, the super-efficiency SBM model also considered the undesirable output, which can calculate the TFP of the WEF nexus in Inner Mongolia cities more objectively and comprehensively. Zhou et al. [45] added carbon dioxide as an undesired output to the DEA model, improving the credibility and guidance of the calculation results of energy efficiency. So, considering the undesirable outputs, the super-efficiency SBM model was used to study the TFP of the WEF nexus in Inner Mongolia cities in this paper.

The Malmquist productivity index is commonly used to measure the change in the TFP of the DMU in two periods. With further research, the methods that make up the TFP are constantly developing. Färe et al. [46,47] proposed that the Malmquist productivity index can be decomposed into two parts, efficiency change (EC) and technical progress change (TC), and efficiency change can be further divided into pure technical efficiency change (PEC) and scale efficiency change (SEC). Moreover, pure technical efficiency change measured the technical efficiency with variable returns to scale, and scale efficiency change measured the technical efficiency between variable returns to scale and constant returns to scale. Reckoning the undesirable output, Luenberger et al. [48] proposed the Luenberger productivity index based on the input distance function. Then Chung [49] put undesirable outputs into the efficiency research framework, utilizing the directional output distance function. Chambers et al. [50] clearly defined Luenberger productivity index, which took into account undesirable outputs. After that, Chung et al. [51] defined the Malmquist–Luenberger (ML) index combining Malmquist productivity index with Luenberger productivity index, which measured the change trend of the TFP considering undesirable output more scientifically and reasonably. Wang et al. [52] utilized the ML index to estimate the environmental TFP of cities in Central China, involving undesirable output into the production process, then the ML index was decomposed into TC and EC. The ML index can effectively analyze the change trend of the TFP and study the reasons for such changes from the four aspects of TC, EC, SEC and PEC. Therefore, there was value in using the ML index to study the change trend of the TFP of the WEF nexus in Inner Mongolia cities in this paper.

The research about the influential factors of the input–output efficiency of the WEF nexus mostly focused on the efficiency of a single resource or two resources. Wang et al. [53] studied the influential factors of environment and energy efficiency in industrial sectors in all provinces of China. Wei et al. [54] studied the influential factors of energy efficiency and mainly considered the influence of factors such as technology innovation, industrial structure, government support, openness to the outside, etc. However, in this paper, we mainly study the influential factors of efficiency from the perspective of the WEF nexus. Based on previous studies, this paper mainly considers the influence of enterprise scale, industrial structure, openness to the outside, government support and other factors.

At present, studies on the TFP of the regional WEF nexus are very few, and research on the influential factors of the TFP are quite rare. Studies on the TFP and influential factors are conductive to the comprehensive discovery of the problems in the efficiency of the regional WEF nexus. The main factors influencing the efficiency of the regional WEF nexus can be found. We selected Inner Mongolia cities as the research area and explored the efficiency between water resources, energy and food subsystems. Based on this, some suggestions will be put forward to achieve better sustainable development of economy, society and ecology in Inner Mongolia cities. The motivation of this paper is to construct an evaluation index system including undesirable output on the basis of understanding the WEF nexus. The weight of each index is calculated by the entropy weight method. Then super-efficiency SBM model is used to measure the TFP of the regional WEF nexus, the ML index is adopted to investigate the change trend of the TFP. Based on this, the Tobit regression is applied to explore and test the main factors that affected the TFP of the regional WEF nexus. The research can provide a theoretical framework and technical support for comprehensive management and sustainable development in Inner Mongolia in the future. The paper is organized as follows. Section 2 introduces the study area. Section 3 describes the methods and data. The main results and discussion are presented in Section 4. Section 5 gives the conclusions of the study.

## 2. Study Area

Inner Mongolia (37°24′–53°23′ N, 97°12′–126°04′ E) is located in the north of China, including nine prefectural cities and three alliances, namely, Hohhot, Baotou, Hulunbuir, Wuhai, Chifeng, Tongliao, Ulaan chal, Baynnur, Ordos, Xing’an, Xilin Gol and Alax (Figure 1). The terrain of Inner Mongolia extends from the northeast to the southwest in a narrow and long shape, which has a vast territory with a total area of 1,183,000 square kilometers, accounting for 12.3% of the land of China. The whole region consists of mountain, hill, plain, desert, river, lake and other landforms, and has a temperate continental climate. Rainfall is rare. Water resources are unevenly distributed in Inner Mongolia. Some cities are rich in water resources, while others are shortage of water resources. Most areas in Inner Mongolia are prone to a shortage of water resources but are abundant in mineral resources. Therefore, it is a typical region for studies on the WEF nexus.

## 3. Methods and Data

### 3.1. Date Sources

This paper used annual panel data ranging from 2007 to 2016. Water resources data such as water consumption were derived from the water resources statistical yearbook of Inner Mongolia. Energy data such as energy consumption were derived from the energy statistical yearbook of Inner Mongolia. Food data such as food consumption were derived from economic and social development bulletins of Inner Mongolia. Economic and social indicators such as GDP, labor force and capital came from the statistical yearbook of Inner Mongolia. Environmental indicators such as waste gas emission, wastewater emission and solid waste emission are from the environmental bulletin of Inner Mongolia and the environment statistical yearbook of Inner Mongolia. Most of these data can be acquired on the official website. Other data can be obtained through field research from government departments of water resources, energy and food.

### 3.2. Establish Evaluation Index System

The construction of the evaluation index system for the TFP of the regional WEF nexus ought to comply with such principles from aspects such as systematicness, dynamics and scientificity. The WEF nexus can be divided into a core association relationship and a peripheral association relationship. The core correlation refers to the correlation between water resource, energy and food. The peripheral correlation refers to the correlation between the ecological system and the social system and the WEF resource system, including the social subsystem, economic subsystem and environmental subsystem [3]. Therefore, we established the evaluation index system from the perspective of input and output. In terms of input indicators, it includes direct input and indirect input indicators. Direct input indicators are the total water consumption, energy consumption and food consumption expenditure of each city in that year. The indirect input indicators are labor force and capital. As one of the core input indicators, labor force is closely connected with regional economic development, so the number of labor forces is selected. Moreover, the facilities and equipment formed by the fixed asset investment help to describe the capital, which promotes the consumption of WEF and the development of the regional economy. Fixed-asset investment is measured by the perpetual inventory method which is shown in Equation (1).
(1)$$K_{t}=I_{t}+K_{t-1}(I-\delta_{t})$$
where $K_{t}$ is the stock of capital in period t, $I_{t}$ is the fixed assets investment and $\delta_{t}$ is the rate of depreciation.

In terms of output indicators, we primarily take into consideration two dimensions: economy and environment. In the aspect of economy, the inputs of water, energy and food were used to further improve the overall economic development in Inner Mongolia cities, so GDP was selected as a desirable output. From the perspective of the environment, the inputs of water, energy and food not only generate GDP, but also produce environmental pollutants such as wastewater, sulfur dioxide, smoke, dust, waste gas, etc. Therefore, waste gas emission, wastewater emission and solid waste emission were selected as the undesirable environmental outputs in this paper. The evaluation index system is shown in Table 1 and the entropy weight method, which can reduce the influence of factors on index weight, was used to weigh the input and output indices. Chen et al. [55] made use of the entropy weight method to determine the weight of each index.

### 3.3. Determine Index Weight

In this paper, different indicators are of different types—the larger the attribute values of some indicators, such as GDP, the better. While the smaller the attribute values of some indicators, such as waste gas emission, the better. In consideration of different attribute values, indicators in this study are divided into positive indicators and negative indicators. Original data should be normalized by Equations (2) and (3) within the range of [0,1].

For the positive indicators:(2)$$x_{i,j}^{t}=\frac{k_{i,j}^{t}-\min k_{i,j}^{t}}{\max k_{i,j}^{t}-\min k_{i,j}^{t}}$$

For the negative indicators:(3)$$x_{i,j}^{t}=\frac{\max k_{i,j}^{t}-k_{i,j}^{t}}{\max k_{i,j}^{t}-\min k_{i,j}^{t}}\;$$

Here, $k_{{}_{i,j}}^{t}$ is the value of variable j of the city i in year t, $\max k_{i,j}^{t}$ represents the maximum value of variables, $\min k_{i,j}^{t}$ represents the minimum value of variables, and $x_{{}_{i,j}}^{t}$ is the standardized value.

After the original data is standardized, the entropy weight method was used to calculate the weight of each indicator. The entropy weight method is an objective weighting method [56] which can reduce the influence of factors on index weight. Furthermore, the entropy of indicator j can be obtained through Equation (4).
(4)$$e_{j}=-k\sum_{i=1}^{m}p_{i,j}\cdot \;\ln p_{i,j}$$

Here, k=1/lnm; $p_{i,j}=x_{i,j}^{t}/\sum_{i=1}^{n}x_{i,j}^{t}$; If $p_{i,j}=0$, then $p_{i,j}\ln p_{i,j}=0$; $0\leq e_{j}\leq 1$.

Finally, the entropy weight of indicator j can be calculated through Equation (5).
(5)$$w_{j}=(1-e_{j})/\sum_{j=1}^{m}1-e_{j}$$

Here, $0\leq w_{j}\leq 1$.

### 3.4. Research Methods

To study the efficiency of the WEF nexus in Inner Mongolia cities, we used the super-efficiency SBM model to calculate the values of the TFP of the WEF nexus by constructing the evaluation index system, which can realize the complete ranking of all DMUs. The evaluation results are more accurate. Based on the calculated results, the ML index was applied to study the change trend of the TFP from a dynamic view. Finally, we made use of the Tobit model to study the influential factors of the TFP of the regional WEF nexus. Based on this, we can put forward policy suggestions to achieve better sustainable development of economy, society and ecology in Inner Mongolia cities.

#### 3.4.1. Efficiency Measurement Model

The DEA model is a new field of interdisciplinary studies on operations research, management science and mathematical economics, which) is generally used to evaluate multiple orientation and multiple decision-making priorities for each DMU. The model is not affected by dimension and subjective factors. Therefore, the DEA method is appropriate for studying and describing efficiency [57]. The DEA models that are utilized the most are the CCR and BCC models [58]. The mathematical expression of the input CCR model is as follows:(6)$$\begin{array}{l} \min\theta \\ s.t.\;\sum_{j=1}^{n}\lambda_{j}X+s^{-}=\theta x_{0},\;\sum_{j=1}^{n}\lambda_{j}Y-s^{+}=y_{0}\\ s^{-}\geq 0,\;s^{+}\geq 0,\;\lambda_{j}\geq 0 \end{array}$$

Here, θ represents technical efficiency of the DMU. X is the input vector. Y is the output vector. $\lambda_{j}$ is weight variable. n is the number of DMUs. j is the DMU. $s^{-}$ and $s^{+}$ are the slack variable of inputs and outputs.

For the case of variable returns to scale, the mathematical expression of the input BCC model is as follows:(7)$$\begin{array}{l} \min\rho \\ s.t.\;\sum_{j=1}^{n}\lambda_{i}X+s^{-}=\rho x_{0},\;\sum_{j=1}^{n}\lambda_{i}Y-s^{+}=y_{0}\\ s^{-}\geq 0,\;s^{+}\geq 0,\;\lambda_{i}\geq 0,\;\sum_{j=1}^{n}\lambda_{j}=1 \end{array}$$

Here, ρ represents the technical efficiency of the DMU.

On the basis of the DEA model, the SBM model has been proposed by Tone [42]. Since the SBM model has no requirements on orientation selection, it can calculate the efficiency of each DMU more accurately. The SBM model is defined as follows:(8)$$\min\rho =\frac{1-\frac{1}{m}\sum_{i=1}^{m}\frac{s_{i0}^{-}}{x_{i0}}}{1+\frac{1}{s}\sum_{r=1}^{s}\frac{s_{r}^{+}}{y_{r0}}}$$
(9)$$s.t.\;x_{0}=X\lambda +s^{-},\;y_{0}=Y\lambda -s^{+}$$
(10)$$\lambda \geq 0,\;s^{-}\geq 0,\;s^{+}\geq 0$$

Here, m is the number of input variables and s is the number of output variables. λ represents the weight vector.

However, the SBM model ignores the impact of undesirable output on efficiency. In the process of production, undesirable outputs are also produced. Therefore, considering undesirable outputs, the SBM model is put forward, which is shown as follows.
(11)$$\min\rho =\frac{1-\frac{1}{m}\times \sum_{i=1}^{m}\frac{s_{i}^{-}}{x_{i_{0}}}}{1+\frac{1}{s_{1}+s_{2}}\times \left(\sum_{r=1}^{s_{1}}\frac{s_{r}^{g}}{y_{r_{0}}^{g}}+\sum_{r=1}^{s_{2}}\frac{s_{r}^{b}}{y_{r_{0}}^{b}}\right)}$$
(12)$$s.t.\;x_{0}=X\lambda +s^{-},\;y_{0}^{g}=Y^{g}\lambda -s^{g},\;y_{0}^{b}=Y^{b}\lambda +s^{b}$$
(13)$$\lambda \geq 0,\;s^{-}\geq 0,\;s^{g}\geq 0,\;s^{b}\geq 0$$

Here, $s_{1}$ is the number of desirable output variables and $s_{2}$ is the number of undesirable output variables. X is the input vector. $Y^{g}$ and $Y^{b}$ are the desirable output vector and undesirable output vector. $s^{-}$, $s^{g}$ and $s^{b}$ are the slack variables of inputs, desirable outputs and undesirable outputs. ρ represents the efficiency of the DMU.

Since the value of each effective DMU calculated by the SBM model is 1, it is impossible to compare and distinguish effective DMUs. In view of this defect, the super-efficiency SBM model was built based on the traditional model, which can compare and sort effective DMUs. Considering undesirable outputs, the super-efficiency SBM model was applied in this study based on the following equation:(14)$$\min\rho =\frac{1+\frac{1}{m}\sum_{i=1}^{m}\frac{s_{i}^{-}}{x_{ik}}}{1-\frac{1}{s_{1}+s_{2}}(\sum_{r=1}^{s_{1}}\frac{s_{r}^{g+}}{y_{rk}^{g}}+\sum_{r=1}^{s_{2}}\frac{s_{t}^{b-}}{y_{rk}^{b}})}$$
(15)$$s.t.\;\sum_{j=1,j\neq k}^{n}X\lambda -s^{-}\leq x_{ik},\;\sum_{j=1,j\neq k}^{n}Y^{g}\lambda +s^{g}\geq y_{{}_{rk}}^{g},\;\sum_{j=1,j\neq k}^{n}Y^{b}\lambda -s^{b}\leq y_{tk}^{b}$$
(16)$$s^{-}>0,\;s^{b}>0,\;s^{g}>0,\;\lambda >0$$
(17)i=1,⋯,m  j=1,⋯,n  r=1,⋯,q

Here, ρ represents the value of the TFP of the DMU. If ρ<1, then the DMU is invalid; If ρ=1, then the DMU is valid; If ρ>1, then the DMU is valid, and a greater ρ represents higher efficiency.

#### 3.4.2. The ML Index

The ML index was adopted to measure the change trend of the TFP of a particular DMU during the study period. The ML index is presented in Equation (18).
(18)$${\mathrm{ML}}^{t,t+1}(x^{t+1},y^{t+1},z^{t+1},x^{t},y^{t},z^{t})={\left[\left(\frac{D^{t}(x^{t+1},y^{t+1},z^{t+1})}{D^{t}(x^{t},y^{t},z^{t})}\right)\times \left(\frac{D^{t+1}(x^{t+1},y^{t+1},z^{t+1})}{D^{t+1}(x^{t},y^{t},z^{t})}\right)\right]}^{1/2}$$

Here, t is time. x, y and z are the input index, desirable output index and undesirable output index. ${\mathrm{ML}}^{t,t+1}$ represents the change in the TFP of the DMU from time t to time t+1. When ${\mathrm{ML}}^{t,t+1}>1$, it shows that the efficiency of the DMU is improved. When ${\mathrm{ML}}^{t,t+1}=1$, it shows that the efficiency of the DMU is static. When ${\mathrm{ML}}^{t,t+1}<1$, it shows that the efficiency of the DMU is reduced.

$D^{t}(x^{t},y^{t},z^{t})$ represents the technical efficiency level in time t compared with the technology conditions in time t. $D^{t}(x^{t+1},y^{t+1},z^{t+1})$ represents the technical efficiency level in time t+1 compared with the technology conditions in time t. $D^{t+1}(x^{t},y^{t},z^{t})$ represents the technical efficiency level in time t compared with the technology conditions in time t+1. $D^{t+1}(x^{t+1},y^{t+1},z^{t+1})$ represents the technical efficiency level in time t+1 compared with the technology conditions in time t+1.

The ML index can be decomposed into technical progress change (TC) and efficiency change (EC), which can effectively reflect the technological development and the management level of the DMU. Specific expressions of TC and EC are shown in Equations (19) and (20).
(19)$$\mathrm{TC}={\left[\left(\frac{D^{t}(x^{t+1},y^{t+1},z^{t+1})}{D^{t+1}(x^{t+1},y^{t+1},z^{t+1})}\right)\times \left(\frac{D^{t}(x^{t},y^{t},z^{t})}{D^{t+1}(x^{t},y^{t},z^{t})}\right)\right]}^{1/2}$$
(20)$$\mathrm{EC}=\left(\frac{D^{t}(x^{t+1},y^{t+1},z^{t+1})}{D^{t}(x^{t},y^{t},z^{t})}\right)$$

Efficiency change can be further resolved into pure technical efficiency change (PEC) and scale efficiency change (SEC). PEC represents the change in technical efficiency under variable returns to scale, and SEC represents the impact of economies of scale on efficiency.

#### 3.4.3. Tobit Model

The Tobit model is known as the limited dependent variable model, which is an econometric model established by Tobin [59]. When this model was proposed, it was mainly used to research the demand of consumer goods. Heckman [60] basically established two steps of the Tobit model, which laid the foundation for the development of Tobit model in the future. And Zhou et al. [61] argued that people have paid extensive attention to the Tobit regression, which is modified into various forms, such as the panel data model, the time series model and the non-parametric model, since the emergence of the Tobit model. The Tobit model is presented in Equation (21).
(21)$$\begin{array}{l} y_{i}^{*}=\beta_{0}+\beta_{1}x_{i1}+\beta_{2}x_{i2}+,\ldots ,+\beta_{j}x_{ij}+\varepsilon_{i,t}\\ y_{i}=\left\{\begin{array}{ll} y_{i}^{*} & \;\mathrm{if}\;y_{i}^{*}>0\\ 0 & \;\mathrm{if}\;y_{i}^{*}\leq 0 \end{array}\right.\;i=1,2,\cdots ,n \end{array}$$

Here, $x_{i,j}$ is the jth explanatory variable for the observation i. $y_{i}$ is the dependent variable. $\beta_{1},\;\beta_{2},\cdots ,\;\beta_{j}$ are the coefficients. $\beta_{0}$ represents the intercept item. $\varepsilon_{i,t}$ represents the stochastic disturbance term.

## 4. Results and Discussion

### 4.1. Analysis of the TFP of the WEF Nexus

We hope that by exploring the TFP, we can seek to minimize inputs undesirable outputs to achieve greater efficiency based on existing desirable output. Therefore, the super-efficiency SBM model was used to calculate the TFP of the WEF nexus in Inner Mongolia cities from 2007 to 2016, and the results are shown in Table 2.

As seen from Table 2, the top three cities with average TFP are Ordos (1.212) Hohhot (1.122) and Xing’an (1.157)—21.2%, 12.2% and 15.7% higher than the optimal level, respectively. These three cities are all economically developed cities in Inner Mongolia with equitable economic structures—Ordos has maintained the highest TFP since 2008. Ordos is an important energy, coal and chemical base area in China and one of the economically robust cities in Inner Mongolia. The city has not only developed economically but has also performed tremendously in ecological development and environmental protection in recent years. Its development is much more rapid than other cities. The average value of the TFP in Tongliao was 0.997, close to 1, which achieved efficiency after 2007, when it recorded an inefficient value of 0.553. The average value of the TFP in Baotou was 0.878, reaching 87.8% of the optimal level. At the same time, the average value of the TFP in Baynnur, Wuhai and Alxa was only 0.374, 0.329 and 0.338, respectively. Therefore, we can conclude that the development of Baynnur, Wuhai and Alxa is still in the extensive mode, the economy is backward, and the notion of environmental protection is scant. In addition, the performance of Wuhai was the worst. The average gap between Wuhai and Ordos was observed to be as much as 88.3%, which showed that development of each city in Inner Mongolia was seriously imbalanced, with the difference between cities being significant. In addition, the average value of the TFP in Inner Mongolia was only 0.715 in the past decade. There is still much room to improve the efficiency of the WEF nexus in Inner Mongolia. Therefore, relevant government departments should prevent the widening of the difference and achieve synchronized development by giving support to the backward areas, coordinating and eliminating the gap between different areas.

Based on the above analysis, we found that there was a great difference between Inner Mongolia cities. Therefore, this paper used k-means clustering analysis to divide Inner Mongolia cities. K-means is one of the clustering algorithms, where k represents the number of categories and means represents the mean value. K-means is an algorithm for clustering data points by means. The K-means algorithm divides similar data points by a pre-set k value and the initial centroid of each category. The optimal clustering results are obtained by means of the iterative optimization after partition [62]. Therefore, the cities in Inner Mongolia were taken as categories, and the average values of the TFP of each city were taken as the means. Then Inner Mongolia cities were grouped into three types of regions based on k-means clustering analysis, namely, the higher efficiency zone, moderate efficiency zone and lower efficiency zone. According to the results, it was found that the efficiency value of the higher efficiency zone was concentrated between 1.000 and 1.250, that of the moderate efficiency zone was concentrated between 0.500 and 1.000, and that of the lower efficiency zone was concentrated between 0.001 and 0.500 (Table 3). The different efficiency zones are marked on the map of Inner Mongolia in different colors (Figure 2).

The line chart of the TFP of each city from 2007 to 2016 is drawn in Figure 3. Since Inner Mongolia cities are differentiated by three types of efficiency zones, we studied the change tendency of each city from 2007 to 2016 from the perspective of efficiency zone. For the higher efficiency zone, the change tendencies of the TFP of Ordos and Tongliao were similar, and both were in the lowest point in 2007 and rose in 2008, and the TFP was maintained at approximately 1.20 and 1.10, respectively, after 2008. As for Hohhot, the values of its TFP fluctuated between 1 and 1.2, throughout the study period. The TFP of Xing’an showed a “V” trend, like Hohhot, and the values were all above 1, in an effective state, throughout the study period. It can be seen from the analysis that the TFP of the higher efficiency zone was effective. The economy and environment both developed harmoniously.

For the moderate efficiency zone, the TFP of each city showed large fluctuations. The TFP of Hulunbuir only was above 1 in 2007 and 2014, declined to its lowest point in 2011, only 0.419, then rose to 1.017 with efficiency in 2014 and continued to fall after 2014. Baotou showed a significant “W” trend. The TFP was below 0.7 in 2009 and 2011 to 2013 and was above 1 in other years. Ulaan chal was above 1 from 2008 to 2010 but plunged in 2011 and remained at approximately 0.4 after 2011, in an ineffective state. The results showed that the development of a moderate efficiency zone has been relatively unstable—economic and environmental progress has waxed and waned.

For the lower efficiency zone, the TFP was expected to be very poor, with the average TFP being below 0.5. Specifically, Baynnur, Alxa and Wuhai were lower than 0.5 over the study period, with the highest being slightly above 0.4. The worst performing city was Wuhai, with a TFP of no more than 0.4 throughout the period, which has always been at a very low level and quite poor. Compared with other cities, the development of the lower efficiency zone is very poor. The same input will produce smaller GDP and larger pollutants for the lower efficiency zone. Therefore, it is imperative to improve the management and technology level of each city in the lower efficiency zone.

By comparing the TFP between higher, moderate, and lower efficiency zones, the difference in TFP in Inner Mongolia cities was very significant—the maximum TFP was 1.234 in Ordos, while the minimum TFP was 0.240 only in Alxa. The difference between the TFP in Inner Mongolia cities was as much as 0.994. On the one hand, the higher efficiency zone has efficiently taken on the policy to help maintain and increase the TFP of its WEF nexus in the past 10 years. On the other hand, there is a smaller effect on the promotion of the overall TFP of the WEF nexus for the lower efficiency zone. The TFP of the WEF nexus still has much room for improvement in most Inner Mongolia cities. In particular, with the improvement in public awareness of environmental protection, environmental problems should be given prominent attention. Therefore, from the overall perspective of the WEF nexus, it will become particularly important to improve the TFP of the WEF nexus through relative resource policies. It is only in this way that Inner Mongolia can realize the balanced development of the regional economy and environment as soon as possible.

### 4.2. Dynamic Analysis of the ML Index

In this paper, the ML index was used to calculate the change trend of the TFP of a particular DMU. The ML index can be decomposed into the technical progress change (TC) and efficiency change (EC) during the study period. EC can be split into pure technical efficiency change (PEC) and scale efficiency change (SEC). Therefore, this paper studied the factors affecting the change trend of the TFP from these four aspects. The results are shown in Table 4 and Table 5.

As seen in Table 4, during the sample period (2007~2016), the average ML value of Inner Mongolia cities was 1.101, increased by 10.1 percentage points, indicating that the TFP of the WEF nexus shows a rising trend and progress has been made in capacity upgrading, structural optimization, energy conservation and emission reduction in Inner Mongolia cities. The average technical progress change increased by 12.9 percentage points and the average efficiency change increased by 2.4 percent. It can be seen that the improvement in the TFP in Inner Mongolia cities was primarily attributable to the growth of technical progress change and efficiency change, but the velocity of technical progress change was faster than efficiency change, which played a critical role. In terms of Inner Mongolia cities, the average ML value in each city was above 1 throughout the study period. The city with the largest average ML value was Xing’an, reaching 1.209, indicating that Xing’an is leading Inner Mongolia in the improvement in the TFP of the WEF nexus, whereas that of Baynnur was the smallest. Technical progress change of other cities was almost above 1 throughout the study duration, while efficiency change was above 1 except for Hulunbuir, Baynnur and Wuhai. Thus, it can be seen that progress has been made in resource utilization, structural optimization, resource saving and emission reduction in Inner Mongolia cities. On an annual basis, from 2008 to 2009, the average ML value increased by 7.5%—average technical progress change increased by 62.1%, while the average efficiency change decreased by 24.5%, which indicates that technical progress change was the main driving force for productivity growth in Inner Mongolia cities. Interestingly, the same situation was observed from 2010 to 2013. Therefore, the TFP was mainly restricted by efficiency change in these years. From 2014 to 2015, the average ML value was only 0.786, and it further decreased by 21.4%—average technical progress change also decreased by 9.7%, with average efficiency change decreasing by 11%. Technical progress change and efficiency change together resulted in the decline in the TFP. In other years, efficiency change and technical progress change jointly promoted the improvement in the TFP in Inner Mongolia cities, where management ability and technological innovation ability have been improved. On the whole, the promotion of the TFP was mainly caused by the growth of technical progress change.

In order to further study the decomposition of the TFP in Inner Mongolia cities, efficiency change (EC) was split into pure technical efficiency change (PEC) and scale efficiency change (SEC). The results are shown in Table 5.

The average efficiency change in Inner Mongolia cities was 1.024, an increase of 2.4%—pure technical efficiency change increased by 1.3% and the scale efficiency change increased by 3.6%. Pure technical efficiency change and scale efficiency change jointly promoted the increase in TFP. From 2010 to 2012, average pure technical efficiency change remained at 1—the growth of which was zero. At this time, average efficiency change was equivalent to the average scale efficiency change, which indicated that efficiency change was mainly due to scale efficiency change. From 2012 to 2013, average efficiency change decreased by 0.2%—average pure technical efficiency change decreased by 7.6%, while average scale efficiency change increased by 11.8%, indicating that the decline in efficiency change was primarily due to the decrease in pure technical efficiency change. It also showed that the comprehensive TFP in each city was mainly restricted by pure technical efficiency change in that year. From 2014 to 2015, average efficiency change decreased by 11%—average pure technical efficiency change decreased by 5.8%, while average scale efficiency change decreased by 5%, indicating that the decline in pure technical efficiency change and scale efficiency change jointly resulted in the decrease in TFP. Overall, the promotion of the TFP was primarily driven by the growth of scale efficiency change.

The average ML values of three efficiency zones in Inner Mongolia from 2007 to 2016 were calculated. Figure 4 shows the line chart of the average ML values of each efficiency zone from 2007 to 2016.

During the whole research period, there has been a decline in the average ML values of three efficiency zones—the higher efficiency zone remained and fluctuated in a small range, above 1 from 2007 to 2014, while it decreased to less than 1 from 2014 to 2016. The average ML values of the moderate and lower efficiency zones fluctuated tremendously, especially the lower efficiency zone. From 2010 to 2011, 2013 to 2014, and 2015 to 2016, the average ML values of the lower efficiency zone went through three significant increases and reached a peak value of 1.397 in 2013 to 2014. The average ML value of the moderate efficiency zone reached the maximum value of 1.462 from 2007 to 2008 and, after that, it showed a significant decline in 2008 to 2012. It then rose marginally to 1.321 in 2013 to 2014. Moreover, the average ML values of moderate and lower efficiency zones were inferior to the higher efficiency zone in six of the ten years, indicating that the difference between Inner Mongolia cities is growing.

### 4.3. Analysis of Influential Factors

In the analysis of influential factors, the TFP calculated by super-efficiency SBM model is the dependent variable. As for independent variables, different scholars have different opinions. In previous research, some scholars already studied the influential factors of efficiency. Wei et al. [54] posited that the main influential factors of efficiency included industrial structure, the influence of government, the degree of opening up and the system—the industrial structure was represented by the value added of the third industry proportion in the gross national product (GNP). The influence of government was expressed by the local financial expenditure proportion in the GNP, and the degree of opening up was also shown by the import and export trade proportion in the GNP. Wang et al. [63] included the enterprise scale in studies on influential efficiency factors, which was expressed by the ratio of the industrial output value to the number of enterprises. Zhao et al. [64] argued that technological innovation, economic structure and the economic development level had important influences on efficiency. Wang et al. [53] stated that the industrial structure, economic development level, technological innovation and market competition were important factors affecting the environment and energy efficiency of industrial sectors in China’s provinces.

In summary, the Tobit model is most suitable for studying the influential factors of the TFP, while, in this paper, independent variables contain the enterprise scale, the output of the third industry, the degree of opening up, government support and the mechanization level from the perspectives of enterprise, economics, government and technology (as shown in Table 6). These factors have direct and indirect effects on the TFP of the WEF nexus.

First, we can set up the Tobit regression model which is expressed as Equation (22).
(22)$${\mathrm{lnTFP}}_{i,t}=\beta_{0}+\beta_{1}{\mathrm{lnES}}_{i,t}+\beta_{2}{\mathrm{lnOTI}}_{i,t}+\beta_{3}{\mathrm{lnDOU}}_{i,t}+\beta_{4}{\mathrm{lnGS}}_{i,t}+\beta_{5}{\mathrm{lnML}}_{i,t}+\varepsilon_{i,t}$$

Here, i is Inner Mongolia city. ${\mathrm{TFP}}_{i,t}$ represents the TFP of the city i in year t. $\beta_{1},\;\beta_{2},\cdots ,\;\beta_{5}$ are the coefficients. $\beta_{0}$ represents the intercept item. $\varepsilon_{i,t}$ represents the stochastic disturbance term. In order to reduce the possible fluctuation influence of the sampled data, logarithmic processing was conducted for all variables in the model and regression analysis was conducted. The Tobit regression results are shown in Table 7. As seen from Table 7, the mechanization level and degree of opening up had remarkably positive effects on the TFP in Inner Mongolia cities at the 95% confidence level. Inner Mongolia has a vast territory and the largest per capita area of cultivated farmland. The improvement in mechanization level is conducive to improving the overall agricultural productivity. The higher the degree of opening up in Inner Mongolia cities, the better the import and export trade will be. Simultaneously, there are more opportunities to introduce advanced technology and outstanding talent from foreign countries to promote their own economic and environmental development as efficiently as possible. Enterprise scale and the output of the third industry had significant negative effects on the TFP in Inner Mongolia cities at the 95% confidence level. Large-scale enterprises are supposed to play an important role in the development of the regional economy and environment. However, the regression results showed otherwise. This denotes that the rapid expansion of enterprise scale does not substantially improve the TFP, diametrically with high pollution and low output. One of the reasons is that enterprises do not pay much attention to developing their technical innovation ability. They, however, only focus on enlarging their scale, with the result of the wasting of social resources. In addition, with the rapid development of the third industry, more resources such as production factors, capital, technology, talents and, etc., flow to the third industry. Finally, government support did not pass the significance test on the promotion of the TFP in Inner Mongolia cities at the 95% confidence level. The financial support of the government should play an important role in guiding and motivating the development of local economy and environment. However, the empirical analysis failed to pass the test, indicating that financial support is not strong enough to promote the regional economy and environmental progress. Du et al. [65] studied the factors affecting the TFP of industrial enterprises and found that the government’s financial support had little influence on the TFP. This is mainly because the government departments have not effectively played a guiding role in improving the efficiency of the largest industrial enterprises. In addition, it may be due to the time-lag of government policies, causing the TFP to not be improved.

## 5. Conclusions

In this paper, the super-efficiency SBM model was applied to calculate the TFP of the regional WEF nexus and the ML index was adopted to study the change trend of the TFP in Inner Mongolia cities from 2007 to 2016 on the basis of understanding the WEF nexus. Tobit regression was used to research the main factors that affected the TFP. This paper, therefore, draws the main conclusions and proposes policy suggestions to improve the TFP in Inner Mongolia cities.

Firstly, for the estimated results based on the super-efficiency SBM model, we can see that the average TFP of Ordos, Hohhot and Xing’an were all higher than the optimal level, while the average TFP of Baynnur, Wuhai and Alxa were only 0.374, 0.329 and 0.338, respectively. The average gap between Wuhai and Ordos was observed to be as much as 88.3%, which showed that there was a serious difference between Inner Mongolia cities regarding the development of economy and environment, with serious polarization and pervasive unbalanced development. According to the average values of the TFP, Inner Mongolia cities were divided into the higher, moderate and lower efficiency zones by k-means clustering analysis. Then, we drew the line chart of the TFP of each city from 2007 to 2016. It can be seen that there was harmonious development between the economy and environment in the higher efficiency zone (containing Ordos, Hohhot, Xing’an, and Tongliao) which was effective during the study period. The development of the moderate efficiency zone (including Hulunbuir, Baotou, Ulaan Chal) has been relatively unstable—the economic and environmental progress waxed and waned. In addition, the lower efficiency zone (including Chifeng, Xilin Gol, Baynnur, Wuhai and Alxa) developed poorly and was still in an ineffective state. Therefore, the lower efficiency zone should take measures to transform industries with high pollution and low output into those with low energy consumption and high output and should learn from the higher efficiency zone and exchange knowledge with these well-developed cities. Moreover, it is imperative for the lower efficiency zone to monitor environmental pollutants such as waste gas, wastewater, etc., to create an environmentally friendly society.

Secondly, for the dynamic analysis of the ML index, it can be seen that the average ML value of Inner Mongolia cities was 1.101, representing an increase of 10.1%, indicating that the change trend of the TFP showed an upward trend as a whole. The average technical progress change increased by 12.9 percentage points and the average efficiency change increased by 2.4 percent. The velocity of technical progress change was faster than that of efficiency change, which illustrated that the growth of the TFP was mainly driven by technical progress change. Then, pure technical efficiency change increased by 1.3% and the scale efficiency change increased by 3.6%. Pure technical efficiency change and scale efficiency change jointly promoted the increase in TFP. Finally, the average ML values of the three efficiency zones in Inner Mongolia from 2007 to 2016 were calculated. The results showed that the average ML values of moderate and lower efficiency zones were inferior to those of the higher efficiency zone in six of the ten years, so it can be seen that the difference between Inner Mongolia cities is growing gradually. Therefore, the government should actively lead the industrial upgrading and phase out high-pollution enterprises in moderate and lower efficiency zones. The government should provide capital, technology and manpower to support the development of the moderate and lower efficiency zones. In addition, point-to-point support can be set up between different efficiency zones.

Finally, based on the analysis of the Tobit regression, the mechanization level and the degree of opening up had significantly positive effects on the TFP in Inner Mongolia cities at the 95% confidence level. That is because the improvement in the mechanization level is conducive to improving the overall agricultural productivity and there are more opportunities to introduce advanced technology and outstanding talent. Enterprise scale and the output of the third industry had negative effects on the TFP at the 95% confidence level. This denoted that the rapid expansion of the enterprise scale does not substantially improve the TFP, diametrically with high pollution and low output. Government support did not have any significant impact on the TFP. It is illustrated that financial support is not strong enough to promote the regional economy and environmental progress. On the one hand, enterprises should adjust their scale appropriately, because too large an enterprise scale may cause the waste of social resources, while too small would not generate a scale effect. Moreover, governments should strengthen support for the moderate and lower efficiency zones to actively promote the transformation of the development model from extensive to intensive. On the other hand, Inner Mongolia cities should also actively strengthen cooperation and exchanges with foreign countries, and introduce more investment in technology-intensive industry. Through these measures, moderate and lower efficiency zones can give play to the advantage of being a late mover. The difference between higher, moderate and lower efficiency zones should be narrowed as quickly as possible.

In this paper, the super-efficiency SBM model combined with the ML index was adopted to calculate the TFP of the WEF nexus in Inner Mongolia cities and its change trend. However, at present, most scholars have considered the efficiency of a single resource or two resources rather than the overall efficiency of the WEF nexus, so this paper holds particular significance. Furthermore, the DEA model is the most common way to calculate the efficiency, but the traditional DEA model cannot effectively deal with the problem of undesired output, and the super-efficiency SBM model makes up for this defect to some extent. Therefore, there is some innovation in the method presented in this paper. In addition, the study on the TFP of the WEF nexus will help to improve the management level of the WEF system in Inner Mongolia in the future, and provide reference for other regions.

## Figures and Tables

**Figure 1 ijerph-16-03051-f001:**
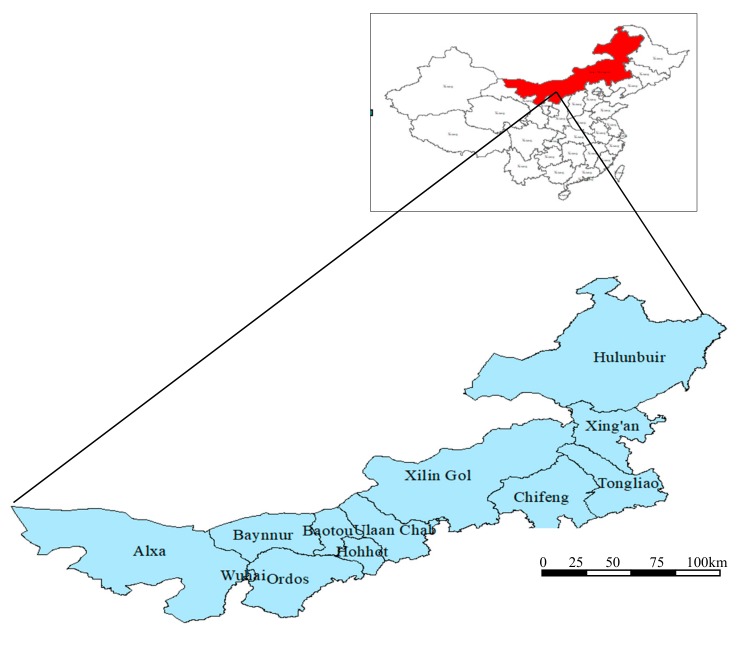
The location of 12 cities in Inner Mongolia, China.

**Figure 2 ijerph-16-03051-f002:**
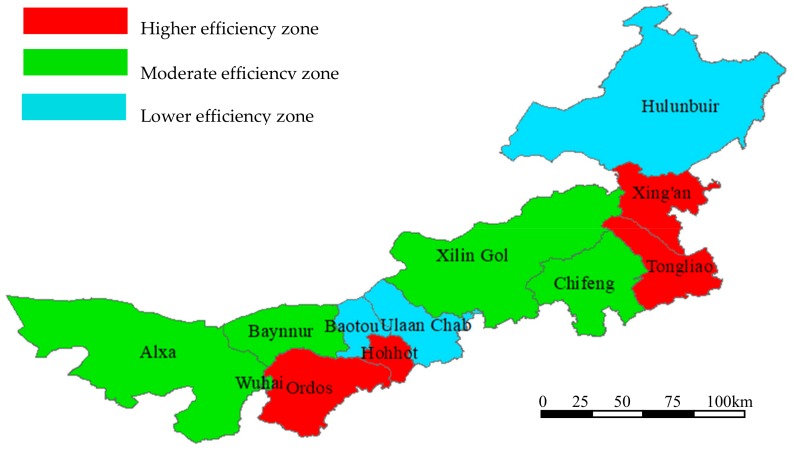
The location of different efficiency zones on the map of Inner Mongolia.

**Figure 3 ijerph-16-03051-f003:**
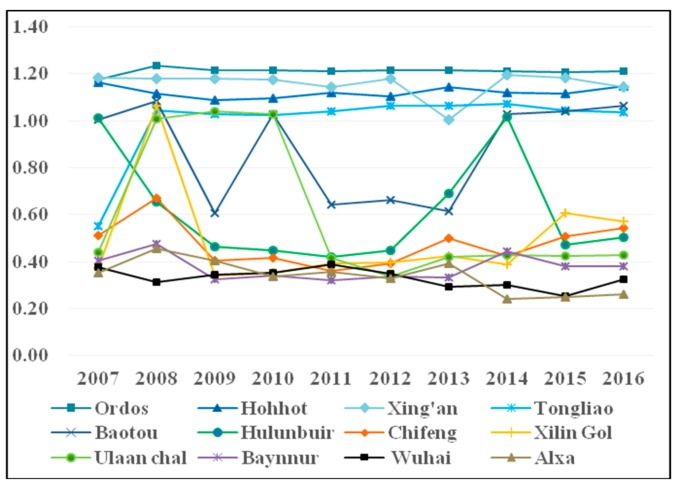
The line chart of the TFP in Inner Mongolia cities from 2007 to 2016.

**Figure 4 ijerph-16-03051-f004:**
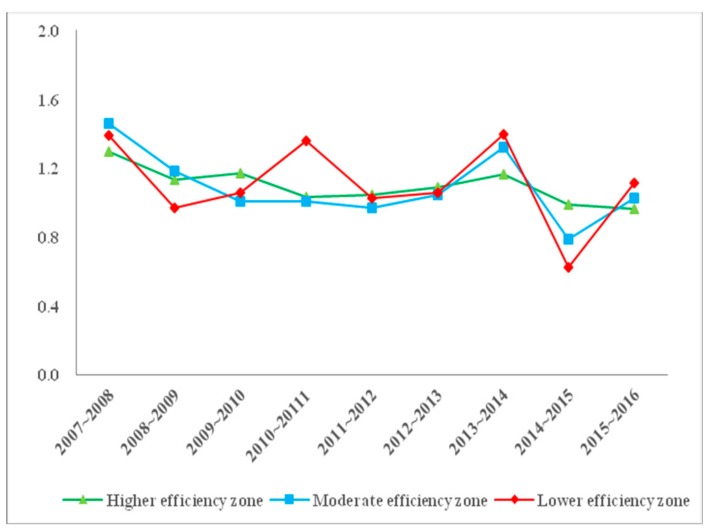
The line chart of average ML values of higher, moderate and lower efficiency zones from 2007 to 2016.

**Table 1 ijerph-16-03051-t001:** Evaluation index system of the total factor productivity (TFP) of the water–energy–food (WEF) nexus in Inner Mongolia cities.

Indicator	First-Class Indicator	Second-Class Indicator	Unit
Input indicators	Water resources	Water consumption	Million cubic meters
	Energy	Energy consumption	10,000 ton of standard coal
	Food	Food consumption	10,000 ton
	Capital	Fixed-asset investment	10,000 yuan
	Labor force	Number of labor force	10,000 persons
Output indicators	Desirable output	GDP	100 million yuan
	Undesirable output	Waste gas emission	Billion cubic meters (BCM)
		Wastewater emission	10,000 ton
		Solid waste emission	10,000 ton

**Table 2 ijerph-16-03051-t002:** The values of the TFP of the WEF nexus in Inner Mongolia cities from 2007 to 2016.

DMU	2007	2008	2009	2010	2011	2012	2013	2014	2015	2016	Average
Ordos	1.176	1.234	1.217	1.217	1.213	1.214	1.216	1.213	1.207	1.210	1.212
Hohhot	1.162	1.117	1.088	1.097	1.120	1.105	1.145	1.120	1.118	1.147	1.122
Xing’an	1.184	1.182	1.180	1.177	1.142	1.182	1.005	1.194	1.184	1.143	1.157
Tongliao	0.553	1.045	1.027	1.026	1.041	1.064	1.065	1.072	1.044	1.037	0.997
Baotou	1.006	1.085	0.605	1.027	0.642	0.662	0.615	1.030	1.040	1.064	0.878
Hulunbuir	1.015	0.657	0.462	0.450	0.419	0.450	0.690	1.017	0.472	0.504	0.614
Ulaan chal	0.440	1.009	1.042	1.029	0.418	0.335	0.419	0.428	0.423	0.428	0.597
Chifeng	0.514	0.669	0.406	0.416	0.360	0.394	0.498	0.424	0.509	0.543	0.473
Xilin Gol	0.373	1.066	0.342	0.353	0.392	0.398	0.423	0.388	0.607	0.569	0.491
Baynnur	0.404	0.475	0.326	0.341	0.322	0.335	0.332	0.444	0.382	0.379	0.374
Wuhai	0.375	0.313	0.346	0.351	0.389	0.348	0.294	0.301	0.253	0.326	0.329
Alxa	0.354	0.456	0.405	0.335	0.355	0.330	0.393	0.240	0.247	0.263	0.338
Average	0.713	0.859	0.704	0.735	0.651	0.651	0.675	0.739	0.707	0.718	0.715

**Table 3 ijerph-16-03051-t003:** The results of the k-means clustering analysis.

Efficiency Zone	Inner Mongolia Cities
Higher efficiency zone (1.000~1.250)	Ordos, Hohhot, Xing’an, and Tongliao
Moderate efficiency zone (0.500~1.000)	Hulunbuir, Baotou, and Ulaan Chal
Lower efficiency zone (0.001~0.500)	Chifeng, Xilin Gol, Baynnur, Wuhai, and Alxa

**Table 4 ijerph-16-03051-t004:** The values of the Malmquist–Luenberger (ML) index, technical progress change (TC) and efficiency change (EC) in Inner Mongolia cities from 2007 to 2016. (DMU = decision-making unit.).

**DMU**	**2007~2008**			**2008~2009**			**2009~2010**			**2010~2011**			**2011~2012**		
	**ML**	**TC**	**EC**	**ML**	**TC**	**EC**	**ML**	**TC**	**EC**	**ML**	**TC**	**EC**	**ML**	**TC**	**EC**
Ordos	1.000	1.000	1.000	1.073	1.073	1.000	1.004	1.004	1.000	1.035	1.035	1.000	1.014	1.014	1.000
Hohhot	0.960	0.960	1.000	1.046	1.046	1.000	1.006	1.006	1.000	0.990	0.990	1.000	1.006	1.006	1.000
Xing’an	1.307	1.307	1.000	1.076	1.778	0.605	1.684	1.020	1.652	1.077	1.677	0.642	1.039	1.008	1.030
Tongliao	1.924	1.064	1.807	1.324	1.324	1.000	0.995	0.995	1.000	1.024	1.024	1.000	1.131	1.131	1.000
Baotou	1.000	1.000	1.000	1.004	1.004	1.000	0.990	0.990	1.000	1.007	1.007	1.000	1.020	1.020	1.000
Hulunbuir	0.995	1.787	0.557	1.359	1.637	0.830	1.029	1.057	0.974	1.568	1.682	0.932	1.074	1.001	1.073
Ulaan chal	1.390	1.051	1.274	1.185	1.185	1.000	0.999	0.999	1.000	0.450	1.076	0.418	0.811	1.012	0.802
Chifeng	1.264	0.971	1.303	1.002	1.651	0.607	1.021	0.996	1.025	1.030	1.192	0.865	1.130	1.032	1.095
Xilin Gol	1.130	0.422	1.681	1.093	3.194	0.342	1.073	1.040	1.031	1.967	1.773	1.109	1.065	1.047	1.017
Baynnur	0.976	0.976	1.000	0.985	1.020	0.326	1.085	1.037	1.047	1.122	1.189	0.944	1.072	1.030	1.041
Wuhai	0.986	0.986	1.000	0.412	1.191	0.346	1.100	1.083	1.016	1.312	1.185	1.107	0.877	0.979	0.896
Alxa	2.585	0.916	2.822	1.346	1.346	1.000	1.003	1.003	1.000	1.363	1.363	1.000	0.984	0.984	1.000
Average	1.293	1.037	1.287	1.075	1.621	0.755	1.082	1.019	1.062	1.162	1.266	0.918	1.019	1.022	0.996
**DMU**	**2012~2013**			**2013~2014**			**2014~2015**			**2015~2016**			**The Average Growth**		
	**ML**	**TC**	**EC**	**ML**	**TC**	**EC**	**ML**	**TC**	**EC**	**ML**	**TC**	**EC**	**ML**	**TC**	**EC**
Ordos	1.014	1.014	1.000	1.019	1.019	1.000	0.985	0.985	1.000	0.996	0.996	1.000	1.016	1.016	1.000
Hohhot	1.063	1.063	1.000	0.993	0.993	1.000	0.988	0.988	1.000	1.029	1.029	1.000	1.009	1.009	1.000
Xing’an	1.244	1.338	0.930	1.642	1.010	1.625	0.998	0.998	1.000	0.810	1.219	0.664	1.209	1.262	1.017
Tongliao	1.038	1.038	1.000	0.998	0.998	1.000	0.985	0.985	1.000	1.013	1.013	1.000	1.159	1.064	1.090
Baotou	1.025	1.025	1.000	1.024	1.024	1.000	1.000	1.000	1.000	0.976	0.976	1.000	1.005	1.005	1.000
Hulunbuir	0.894	1.031	0.867	1.364	1.078	1.563	0.438	0.928	0.472	1.031	0.966	1.068	1.084	1.241	0.903
Ulaan chal	1.221	0.976	1.251	1.575	1.542	1.021	0.929	0.939	0.989	1.062	1.050	1.012	1.180	1.092	1.085
Chifeng	1.951	1.162	1.540	0.899	0.899	1.000	0.474	0.931	0.509	1.060	1.048	1.966	1.092	1.098	1.101
Xilin Gol	1.019	0.959	1.063	1.548	1.689	0.917	0.908	0.581	1.564	0.962	1.025	0.939	1.196	1.303	1.074
Baynnur	1.018	1.029	0.990	2.142	1.600	1.339	0.548	0.637	0.861	1.006	1.014	0.992	1.106	1.281	0.949
Wuhai	0.795	0.942	0.844	1.841	1.129	1.402	0.240	0.949	0.253	1.440	1.046	1.959	1.001	1.107	0.980
Alxa	0.495	1.004	0.493	0.533	1.095	0.487	0.946	0.917	1.032	1.093	1.029	1.062	1.150	1.073	1.099
Average	1.065	1.048	0.998	1.298	1.173	1.113	0.786	0.903	0.890	1.040	1.034	1.138	1.101	1.129	1.024

**Table 5 ijerph-16-03051-t005:** The values of EC, pure technical efficiency change (PEC) and scale efficiency change (SEC) in Inner Mongolia cities from 2007 to 2016.

**DMU**	**2007~2008**			**2008~2009**			**2009~2010**			**2010~2011**			**2011~2012**		
	**EC**	**PEC**	**SEC**	**EC**	**PEC**	**SEC**	**EC**	**PEC**	**SEC**	**EC**	**PEC**	**SEC**	**EC**	**PEC**	**SEC**
Ordos	1.000	1.000	1.000	1.000	1.000	1.000	1.000	1.000	1.000	1.000	1.000	1.000	1.000	1.000	1.000
Hohhot	1.000	1.000	1.000	1.000	1.000	1.000	1.000	1.000	1.000	1.000	1.000	1.000	1.000	1.000	1.000
Xing’an	1.000	1.000	1.000	0.605	1.000	0.605	1.652	1.000	1.652	0.642	1.000	0.642	1.030	1.000	1.030
Tongliao	1.807	1.000	1.807	1.000	1.000	1.000	1.000	1.000	1.000	1.000	1.000	1.000	1.000	1.000	1.000
Baotou	1.000	1.000	1.000	1.000	1.000	1.000	1.000	1.000	1.000	1.000	1.000	1.000	1.000	1.000	1.000
Hulunbuir	0.557	0.644	0.864	0.830	0.963	0.862	0.974	1.613	0.604	0.932	1.000	0.932	1.073	1.000	1.073
Ulaan chal	1.274	1.761	1.291	1.000	1.000	1.000	1.000	1.000	1.000	0.418	1.000	0.418	0.802	1.000	0.802
Chifeng	1.303	1.000	1.303	0.607	1.000	0.607	1.025	1.000	1.025	0.865	1.000	0.865	1.095	1.000	1.095
Xilin Gol	1.681	1.000	1.681	0.342	1.000	0.342	1.031	1.000	1.031	1.109	1.000	1.109	1.017	1.000	1.017
Baynnur	1.000	1.000	1.000	0.326	1.000	0.326	1.047	1.000	1.047	0.944	1.000	0.944	1.041	1.000	1.041
Wuhai	1.000	1.000	1.000	0.346	1.000	0.346	1.016	1.000	1.016	1.107	1.000	1.107	0.896	1.000	0.896
Alxa	2.822	1.000	2.822	1.000	1.000	1.000	1.000	1.000	1.000	1.000	1.000	1.000	1.000	1.000	1.000
Average	1.287	1.034	1.314	0.755	0.997	0.757	1.062	1.051	1.031	0.918	1.000	0.918	0.996	1.000	0.996
**DMU**	**2012~2013**			**2013~2014**			**2014~2015**			**2015~2016**			**The Average Growth**		
	**EC**	**PEC**	**SEC**	**EC**	**PEC**	**SEC**	**EC**	**PEC**	**SEC**	**EC**	**PEC**	**SEC**	**EC**	**PEC**	**SEC**
Ordos	1.000	1.000	1.000	1.000	1.000	1.000	1.000	1.000	1.000	1.000	1.000	1.000	1.000	1.000	1.000
Hohhot	1.000	1.000	1.000	1.000	1.000	1.000	1.000	1.000	1.000	1.000	1.000	1.000	1.000	1.000	1.000
Xing’an	0.930	1.000	0.930	1.625	1.000	1.625	1.000	1.000	1.000	0.664	1.000	0.664	1.017	1.000	1.017
Tongliao	1.000	1.000	1.000	1.000	1.000	1.000	1.000	1.000	1.000	1.000	1.000	1.000	1.090	1.000	1.090
Baotou	1.000	1.000	1.000	1.000	1.000	1.000	1.000	1.000	1.000	1.000	1.000	1.000	1.000	1.000	1.000
Hulunbuir	0.867	0.577	1.504	1.563	1.733	1.279	0.472	0.603	0.783	1.068	1.658	0.644	0.903	1.088	0.949
Ulaan chal	1.251	1.000	1.251	1.021	1.000	1.021	0.989	0.706	1.400	1.012	1.416	1.184	1.085	1.098	1.041
Chifeng	1.540	1.000	1.540	1.000	1.000	1.000	0.509	1.000	0.509	1.966	1.000	1.966	1.101	1.000	1.101
Xilin Gol	1.063	1.000	1.063	0.917	1.000	0.917	1.564	1.000	1.564	0.939	1.000	0.939	1.074	1.000	1.074
Baynnur	0.990	1.000	0.990	1.339	1.000	1.339	0.861	1.000	0.861	0.992	1.000	0.992	0.949	1.000	0.949
Wuhai	0.844	0.514	1.643	1.402	1.246	1.748	0.253	1.000	0.253	1.959	1.000	1.959	0.980	0.973	1.107
Alxa	0.493	1.000	0.493	0.487	1.000	0.487	1.032	1.000	1.032	1.062	1.000	1.062	1.099	1.000	1.099
Average	0.998	0.924	1.118	1.113	1.081	1.118	0.890	0.942	0.950	1.138	1.090	1.117	1.024	1.013	1.036

**Table 6 ijerph-16-03051-t006:** The explanatory variable, abbreviation and remarks of influential factors of the TFP.

Variable	Explanatory Variable	Abbreviation	Remarks
The dependent variable	Total factor productivity	TFP	Super-efficiency SBM value
The independent variables	Enterprises scale	ES	The number of workers
	The output of the third industry	OTI	Total output of the third industry/GNP
	Degree of opening up	DOU	Total imports and exports/GNP
	Government support	GS	Total financial expenditures of WEF/GNP
	Mechanization level	ML	The number of mechanical facilities and equipment

**Table 7 ijerph-16-03051-t007:** The results of Tobit regressions.

Explanatory Variable	B	Standard Error	Beta	*t*	Significance
Constant	1.833	0.310		5.909	0.000
ES	−2.144	0.641	−0.647	−3.348	0.002
OTI	−1.812	0.498	−0.529	−3.637	0.001
DOU	3.894	0.957	−0.502	4.069	0.000
GS	1.097	1.032	0.191	1.062	0.297
ML	0.858	0.177	0.636	4.835	0.000

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
