# Peer review of "Research on Total Factor Productivity and Influential Factors of the Regional Water–Energy–Food Nexus: A Case Study on Inner Mongolia, China"

_ijerph, 2019, doi:10.3390/ijerph16173051_

Round 1

Reviewer 1 Report

Major Comments:

The paper present an interesting technique to study the trend in Water-Energy-Food (WEF) productivity in 12 cities. However, authors could strength the paper by bringing the discussion of why and how the techniques are important for WEF early in the manuscript. Also, authors need to clarify the contribution of this work. The suggestion to improve TFP are a very good addition and provides an informative and insightful message.

·        Although it raised the importance of the NEXUS approach, authors didn't discuss the unique methodology and approach used to address NEXUS issues for the study area.

·        It would be interesting to see how the identified influential factors for WEF Nexus are different or similar from the other studies.

·        Authors used different techniques and failed to provide justification of the relevance. In addition to its popularity, authors need to consider if DEA is appropriate for all the data and provide how it is relevant to their work.

·        The paper has important messages and should be of great interest to the readers. However, some of the results are not well presented (Line 290- 330).

·        Redundancy i.e Table 1 and Figure 2 and 3. Also, it is not clear if the authors used the average values for k-mean clustering.  Need to explain how the range/numbers are defined.

·        Authors have examined technical and efficiency changes in detail. It would also be interesting to see the impact of other factors.

·        This paper has numerous grammar and language issues, which need to be addressed (for example Line 190, 207, 357…..)

Minor Comments:

·        Line 22 and 23: Provide specific result.

·        Line 88: When using an acronym for the first time, it must be spelled out i.e DEA .

·        Line 188-195 & 202-209: The assumption made should be discussed clearly and early in the manuscript.

·        Line 190: “my personal understanding”- use words that better described your opinion

·        Line 193: Readers would like to know why labor force is chosen rather than capital as major indirect input indicators.

·        Line 207: delete “and”

·        Line 202: Provide justifications of why economy and environment. Discussion of how social dimensions affect the two dimensions would also be beneficiary.

·        Line 214: Provide justification for the choice of normalization method .

·        Line 231: authors need to provide how the those models are relevant to their work.

·        Line 320-324: Is the average TFP value used for k-mean clustering?  Also, need to explain how the range/number is defined.

·        Line 356-357: Please consider revising repeated phrases i.e on the other hand...

·        Line 365: Location of Table 4 and 5?

·        Line 484: Was there ever a mention of why financial support is not strong to promote regional economy? Author should provide reasoning of why financial support is not strong to promote regional economy. This would provide more weight to this proclamation.

Author Response

    Thank you very much for your valuable comments on our paper. I have provided my response to your comments in PDF file in detail and uploaded it. My heartfelt thanks to you again.

Reviewer 2 Report

I found the paper to be well-written it was not so clear in terms of the theoretical contribution or the methods used. With some reworking to enhance the engagement with theory to show how this work contributes and with more methodological rigor, and a more refined analysis this paper could be published. For this reason, I can make some specific improvement suggestions:

The literature review needs to be better integrated into the paper. What is the purpose of the literature review? How are the topics covered in the literature review justified as relevant? I suggest adding a statement of purpose for the literature review as well as an outline of themes/topics with a justification for each. The literature review would also benefit from subheadings.

The discussion and conclusions do not provide enough detail. Also, the discussion section could benefit from an engagement with the literature that addresses the novelty of the approach, the significance of integrating models and how it might contribute to broader knowledge in the field.

While I suggest these relatively minor revisions of the article, I remain excited about

the manuscript’s contribution to IJERPH. I hope the authors receive this review and strengthen the manuscript, and I look forward to seeing it in publication.

Author Response

    Thank you very much for your valuable comments on our paper. I have provided a point-by-point response to your comments in PDF file and uploaded it. My heartfelt thanks to you again
